# A novel art of continuous noninvasive blood pressure measurement

Jürgen Fortin [1✉], Dorothea E. Rogge[2], Christian Fellner[1], Doris Flotzinger[1], Julian Grond[1], Katja Lerche[1] & Bernd Saugel [2,3]

Wearable sensors to continuously measure blood pressure and derived cardiovascular variables have the potential to revolutionize patient monitoring. Current wearable methods analyzing time components (e.g., pulse transit time) still lack clinical accuracy, whereas existing technologies for direct blood pressure measurement are too bulky. Here we present an innovative art of continuous noninvasive hemodynamic monitoring (CNAP2GO). It directly measures blood pressure by using a volume control technique and could be used for small wearable sensors integrated in a finger-ring. As a software prototype, CNAP2GO showed excellent blood pressure measurement performance in comparison with invasive reference measurements in 46 patients having surgery. The resulting pulsatile blood pressure signal carries information to derive cardiac output and other hemodynamic variables. We show that CNAP2GO can self-calibrate and be miniaturized for wearable approaches. CNAP2GO potentially constitutes the breakthrough for wearable sensors for blood pressure and flow monitoring in both ambulatory and in-hospital clinical settings.

[1] CNSystems Medizintechnik GmbH, Reininghausstrasse 13, 8020 Graz, Austria. [2] Department of Anesthesiology, Center of Anesthesiology and Intensive Care Medicine, University Medical Center Hamburg-Eppendorf, Martinistrasse 52, 20246 Hamburg, Germany. [3] Outcomes Research Consortium, Cleveland, OH, USA. ✉email: juergen.fortin@cnsystems.com

The digital revolution is about to bring innovations to physiologic monitoring[1]. Innovative wearable sensors for monitoring of cardiovascular dynamics have the potential to revolutionize the monitoring of vital signs and their response to therapy in both ambulatory and in-hospital clinical settings. Continuous real-time monitoring of cardiovascular dynamics—including arterial blood pressure (BP) and advanced hemodynamic variables that are derived from its waveform (stroke volume, cardiac output (CO), and dynamic cardiac preload variables)—is a mainstay of patient care in perioperative and intensive care medicine but, for the most part, still requires invasive or stationary noninvasive sensors[2,3]. With the availability of noninvasive and wearable sensors, advanced cardiovascular monitoring may become part of patient surveillance on normal wards[4] and even outside the hospital[5]. Upgrading patient monitoring capabilities with wearable solutions designed for normal wards may help avoid intensive care unit overload (as seen during the COVID-19 crisis)[6].

For these clinical applications, wearable sensors have to fulfill all regulatory and clinical demands of medical-grade devices—just as their stationary counterparts. Challenges for future developments in the field of wearable miniaturized monitoring sensors will be to provide measurements with clinically acceptable accuracy, precision, and reliability and to ensure clinical usability and patient safety. Ideally, miniaturized monitoring sensors measure in the background without user interaction. At the moment, wearable sensors for continuous BP monitoring still show poor measurement performance[7] and have, therefore, not been adopted into clinical practice.

Most current wearable BP monitoring systems estimate BP based on time information. Pulse time interval or pulse velocity methods use a proximal and a distal sensor for the measurement of the pulse transit time or pulse arrival time[8]. Other time-based methods for BP estimation use sensors that decompose the pulse into a forward and a backward wave to analyze their time differences[9,10]. There are also attempts to use photo-plethysmographic (PPG)[11] or piezoelectric[12] pulse detection for the estimation of BP based on amplitude and time information. However, a direct translation of the pulse signal—which is a surrogate for volume changes—into continuous BP is challenging because of confounding effects of the cardiovascular, respiratory, and autonomic nervous systems. By enhancing model complexity (e.g., using machine learning and neural network tools), the measurement performance of these approaches may be improved[11,13,14]. Although their sensor hardware is very simple, significant work is still needed to achieve clinically acceptable performance[8,11,15,16]. The difficulty with time-based approaches for wearable BP measurement devices seems to be that the variables of the mathematical models need to change along with changes in vasomotor activity of the vascular smooth muscles which are determined by the autonomic nervous system. All of these cuff-less attempts of continuous BP monitoring require frequent re-calibration, for example using cuff-based sphygmomanometers, rendering them unfeasible for wearable wireless monitoring[17,18].

Other wearable approaches based on the oscillometric principle provide BP values intermittently. For example, the HeartGuide (Omron, Kyoto, Japan) is a wrist-worn watch-type oscillometric BP monitoring device[19]. Another innovative approach measures oscillometric curves by pressing a finger on a cell phone screen[20,21]. While this research shows that oscillometric BP measurement at the finger is feasible, these methods need user interaction (keeping the sensor at heart level or pressing a finger on the screen) and automated continuous BP monitoring (including monitoring during night sleep) is not possible.

Stationary finger cuff devices have been successfully introduced for direct continuous noninvasive BP monitoring in various medical fields, especially in critical care and anesthesiology[22]. These devices are based on the vascular unloading technique (VUT), also referred to as volume clamp or Peňáz method[23–25]. The VUT measures BP and derives the complete pulsatile BP waveform by controlling fast inflation and deflation of a finger cuff in combination with PPG. As with time-based technologies, the VUT is influenced by changes in vasomotor activity; different devices have found differing methods to detect and compensate for changes in vasomotor tone[26–28]. VUT devices do not necessarily need calibration for proper functioning. Instead, the Finapres (Ohmeda Monitoring Systems, Englewood, CO, USA) and successor devices use a heart level sensor and/or a transfer function to provide estimates of brachial BP values[29,30]. CNAP devices (CNSystems Medizintechnik, Graz, Austria) have a different approach: they routinely use an upper arm cuff for orthostatic/heart level correction and for calibration to brachial BP level[31]. A transfer function approach is also used as a temporary correction until the calibration is performed or if the user does not want to use the upper arm cuff.

Existing VUT systems are, however, too bulky to be used for wearable monitoring devices because the systems consist of mechanical elements like pumps, valves, and air hoses to follow the pulsatile nature of BP with high fidelity.

It was generally believed that, for correct BP measurement based on VUT, a constant (or clamped) blood volume in the finger over the complete period of pulsation and therefore full vascular unloading of the blood vessels is essential. Whether this is indeed physiologically or metrologically necessary was never examined.

The basic idea behind our novel art of continuous noninvasive BP measurement, CNAP2GO, is to perform the blood volume control substantially more slowly to be able to use slow-moving and, therefore, small-scaled hardware. Most importantly, such hardware requires neither a pump nor a valve. The basic principle of the CNAP2GO method is the volume control technique (VCT). In contrast to the VUT, which keeps the blood volume in the finger artery constant on a millisecond basis to accurately follow the full cardiac cycle, the VCT keeps the volume constant on a time scale of heartbeats. Blood volume is allowed to oscillate over heartbeats, balancing inflow and outflow of blood in the finger artery over each heart cycle. Controlling blood volume by VCT without fast oscillations poses a non-trivial control engineering problem, which is solved using a set of signal processing algorithms. Most importantly, as we will demonstrate, this method is resistant to changes in vasomotor tone.

In this paper, we first describe the elements of CNAP2GO and how they interact to derive a continuous BP waveform from the directly measured mean BP (mBP) and the PPG signal. We show that the BP waveform can be used to estimate advanced hemodynamic variables. The VCT was implemented in software and placed on the hardware of an existing VUT-device that emulates slow VCT. The accuracy of the CNAP2GO software prototype was clinically validated in comparison with invasive BP measurements obtained using a radial arterial catheter in patients having surgery with general anesthesia. The CNAP2GO method can, in the future, be used for small wearable sensors integrated in a finger-ring utilizing a light transmitter (LED) and a receiver (photodiode). The contact pressure of LED and photodiode can be modified by an actuator changing the ring diameter with a rate-of-change as fast (or rather: as slow) as mBP.

A wearable approach of an actuator is demonstrated and its feasibility is shown. This actuator is able to produce a typical oscillometric curve measurement including its envelope derived from the PPG signal $v(t)$, which is the basis for VCT. Using this envelope, the method can automatically self-calibrate to oscillometric systolic BP (sBP) and diastolic BP (dBP). For a clinically

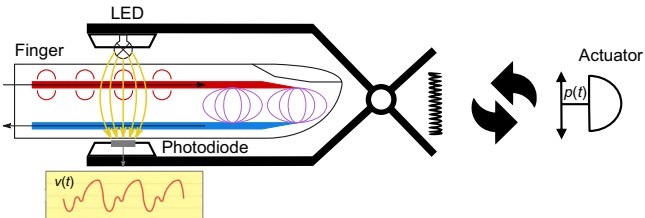

**Fig. 1 Actuator within CNAP2GO.** Within CNAP2GO, the element for constant pressure coupling is a simple, slow-moving actuator that can vary the contact pressure $p(t)$ of the light elements (light-emitting diode and photodiode) as fast as mean blood pressure may change. $v(t)$: light signal.

meaningful continuous BP signal, an integrated heart level sensor utilizing an electronic XYZ-axis accelerometer and gyroscope in addition to a suitable transfer function will be applied. As user interaction is not required, CNAP2GO may be used in a broad range of medical applications.

## Results

**Basic CNAP2GO considerations**. The CNAP2GO sensor is a PPG probe which comprises an LED and a photodiode. The contact pressure of these light elements to the skin can be modified. Even simple PPG probes (e.g., pulsoxymeter) have elements that couple the light components to the body with pressure: this is usually a spring, Velcro, or another simple clamp mechanism. A constant pressure (around 20–30 mmHg) ensures good light coupling. Within CNAP2GO, a simple actuator can vary the contact pressure of the light elements as fast as mBP may change (Fig. 1). As a consequence, the system wins an additional degree of freedom which is needed for continuous BP measurement as it can determine the oscillometric curve.

The basic idea of varying contact pressure is also implemented in all existing devices based on the VUT. When using the VUT, the speed and amplitude of the changes in contact pressure must match pulsatile arterial BP, requiring pressure changes of 30–100 mmHg per systolic upstroke of the beat to ensure high-fidelity pressure waveforms. For example, the pressure system of the CNAP devices can apply pressure changes with more than 1500 mmHg/s.

**CNAP2GO block diagram and signal flow**. The CNAP2GO contact pressure is determined by a closed-loop control system using the PPG signal and its components as input signals. Figure 2 shows the block diagram and the signal flow of CNAP2GO. An LED-light ($\lambda = 890$ nm) is directed through a finger and light absorption is measured by a photodiode with corresponding wavelength characteristics. The resulting PPG signal $v(t)$ is an inverse surrogate for blood volume changes inside the finger artery: Rising blood volume increases absorption and therefore decreases the amount of light transmitted through the finger.

The contact pressure $p_c(t)$ of the light elements can be altered by the actuator and is measured by a pressure gauge. $v(t)$ is led into digital filters producing $v_{\text{Pulse}}(t)$, $v_{\text{Rhythm}}(t)$, and $v_{\text{VCT}}(t)$. The signal components $v_{\text{Rhythm}}(t)$ and $v_{\text{VCT}}(t)$ are used to control $p_c(t)$, whereas $v_{\text{Pulse}}(t)$ together with $p_c(t)$ are needed for the calculation of the pulsatile BP signal $p_{\text{C2G}}(t)$. The control loop between the controller and the pressure is closed using antiresonance elements. Note that $v(t)$ and its components are dimensionless digital signals representing a surrogate of blood volume in the finger.

**Finding the initial setpoint**. Closing the CNAP2GO control loop requires an initial open-loop phase to determine the initial value

of mBP by altering the contact pressure and detecting the corresponding PPG pulse-amplitudes from the signal $v_{\text{Pulse}}(t)$. Figure 3 shows how a typical oscillometric curve including its envelope is obtained. Figure 3a shows the resulting $v(t)$ when a pressure ramp $p_c(t)$ is applied. The $p$-$v$ transfer function corresponds to an S-shaped arcus tangent with superimposed pulses[32]. The upper asymptote of $v(t)$, where $p_c(t)$ is far above sBP, corresponds to the maximum light that can be detected from the finger when all blood is squeezed out and no pulses can be detected. The lower asymptote is at $p_c(t) = 0$ where no deformation of the finger and its artery occurs (although this is a theoretical point of measurement because a minimum contact pressure is needed for proper light coupling). The amplitudes of the $v(t)$-pulses are derived from the high-pass filtered $v_{\text{Pulse}}(t)$. The amplitudes of $v_{\text{Pulse}}(t)$ are fitted using a Gaussian-style envelope curve, as can be seen in Fig. 3b. The peak of this envelope indicates mBP according to the maximum oscillation rule[33], a principle which also holds true when using PPG signals[34–36]. After oscillometric envelope determination, the system applies mBP as initial contact pressure $p_c(t)$ and stores this value as the starting setpoint $P_0$ and its PPG companion $v_0$. For calibration purposes, sBP and dBP can be determined.

While this open-loop phase is very similar to the basic oscillometric principle, as well as to standard VUT, the closed-loop of CNAP2GO differs in almost all fundamental aspects.

**Tracking mBP**. After the closed-loop phase, the mean cuff pressure $p_C(t)$ is located on the inflection point of the S-shaped $p$-$v$ transfer function. As shown in Fig. 4a, the ideal pulsatile $v_{\text{Pulse}}(t)$ (light blue trace) has maximum amplitude and the integral of $v_{\text{Pulse}}(t)$ over a full beat is zero. Thus, the main condition of the VCT, which keeps blood volume in the finger artery balanced over a heart cycle, is fulfilled.

As soon as BP or vasomotor changes occur, the situation becomes different. If $p_C(t)$ is lower than mBP ($p_<$), $v_{\text{Pulse}}(t)$ has a lower amplitude (purple trace) and differs in pulse shape: the pulses are fat. In contrast, if $p_C(t)$ is higher than mBP ($p_>$), $v_{\text{Pulse}}(t)$ has also a lower amplitude, but the pulses are spiky (dark blue trace).

Figure 4b shows how vasoconstriction influences the $p$-$v$ transfer function: while the upper asymptote for $p_C(t) \gg$ sBP remains unchanged, the lower asymptote for $p_C(t) \ll$ dBP shows a higher $v(t)$ because finger blood volume decreases during vasoconstriction. This results in a setpoint lower than the inflection point and a change from $v_0(t)$ to a smaller but fat $v_1(t)$ with the negative half-wave of the fat pulse being greater than the positive half-wave. On the other hand, vasodilation increases the setpoint over the inflection point, thus producing a spiky signal.

For both fat and spiky pulses, the integral of the pulsatile volume signal over a beat is no longer equal to zero. The CNAP2GO controller is responsible for restoring this condition of balanced volume control: in particular, the setpoint $p_0$ and resulting $p_C(t)$ need to be altered after each heartbeat until balance is achieved (Fig. 4c). A proportional-integral (PI) control approach results in an adaptive formula for setpoint correction as can be seen in the methods section below. The summation of the I (integral) part of the control approach is the memory for the setpoints $p_{1,2,...,n}$ that allows for the reconstruction of long-term BP information (see also the Methods section for details).

This mechanism not only works for vasomotor changes but also for pure mBP changes. An increase in mBP—which is a shift of the $p$-$v$ transfer function to the right—tends to move the setpoint towards the lower asymptote of the $p$-$v$ transfer function. The integral over the beat will be negative resulting in an increase

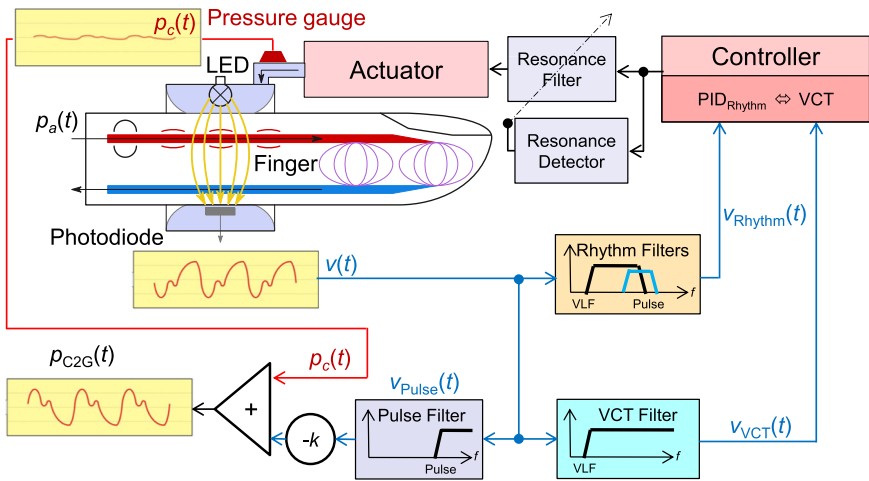

**Fig. 2 Block diagram and signal flow of CNAP2GO.** An LED-light is directed through a finger and its absorption is measured by a photodiode depending on arterial pressure $p_a(t)$. The resulting photo-plethysmographic signal $v(t)$ is led into digital filters producing $v_{Pulse}(t)$, $v_{Rhythm}(t)$, and $v_{VCT}(t)$. Signal components $v_{Rhythm}(t)$ and $v_{VCT}(t)$ are used to control the pressure $p_c(t)$, whereas $v_{Pulse}(t)$ together with $p_c(t)$ are needed for the calculation of the pulsatile blood pressure signal $p_{C2G}(t)$. The control loop using a proportional-integral-differential controller $PID_{Rhythm}$ and the volume control technique (VCT) is closed using antiresonance elements. $f$: frequency, VLF: very low-frequency cutoff, $k$: amplification factor.

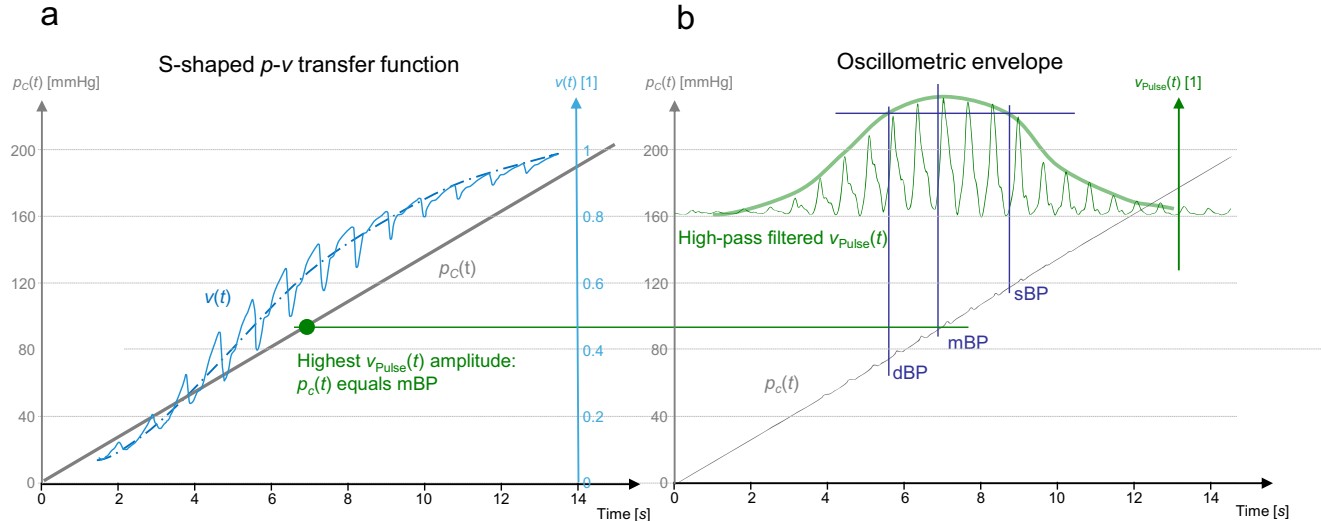

**Fig. 3 Depiction of a typical oscillometric curve including its envelope.** **a** The resulting $v(t)$ (solid blue line) when a pressure ramp $p_c(t)$ is applied (ideal pressure ramp shown as the gray line). Averaged signal (dashed-dotted line). The green dot indicates the pressure where the light pulsations are highest. **b** Amplitudes of $v_{Pulse}(t)$ (thin solid line) are fitted using a Gaussian-style envelope curve (thick solid line). Systolic (sBP), diastolic (dBP), and mean blood pressure (mBP) can be derived from those curves. The gray line shows the measured pressure signal $p_c(t)$.

of the setpoint $p_n$. Vasodilation shifts the S-curve in the opposite way and VCT decreases $p_n$.

**Calibration.** CNAP2GO's $p_C(t)$ tracks mBP and, with a simple calculation, the pulsatile nature of BP can be superimposed using $v_{Pulse}(t)$:

$$p_{C2G}(t) = p_C(t) + k \cdot v_{Pulse}(t) \text{ whereas } k = \frac{sBP_{init} - dBP_{init}}{v_{sys} - v_{dia}}$$

(1)

where $sBP_{init}$ and $dBP_{init}$ are calibration values and $v_{sys}$ and $v_{dia}$ are maxima and minima of $v_{Pulse}(t)$ during the initial phase.

The calibration values $sBP_{init}$ and $dBP_{init}$ can be obtained as external (e.g., brachial) values or from the initial phase (see Fig. 3). $sBP_{init}$ and $dBP_{init}$ retrieved from finger oscillometry will need a heart level correction sensor as well as an adapted transfer

function known from literature[29,37]. The same is true for finger oscillometric algorithms including its parametrization[38], although a detailed development work including validation will be needed.

**CNAP2GO prototype implementation.** We implemented the CNAP2GO mechanism as a software prototype into existing hardware. A commercially available CNAP Monitor HD was adapted via software (see Methods section) to turn its VUT into VCT (CNAP2GO). Although the CNAP hardware in principle allows for the fast pulsatile BP changes as necessary for VUT, the speed of pressure changes was inherently limited by the software design of CNAP2GO as described in this article.

There were three reasons for this approach: First, the hardware of the CNAP Monitor HD has international regulatory approval for use in patients. Second, the experiments could be performed prior to expensive investments into miniaturization. And finally,

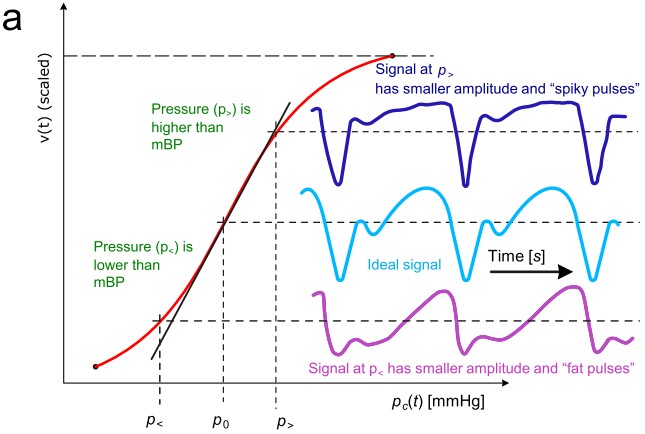

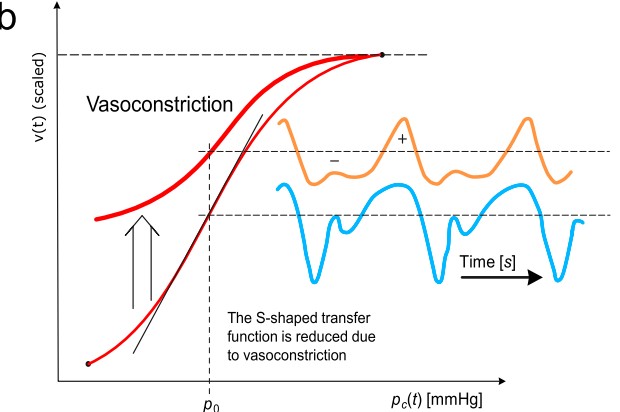

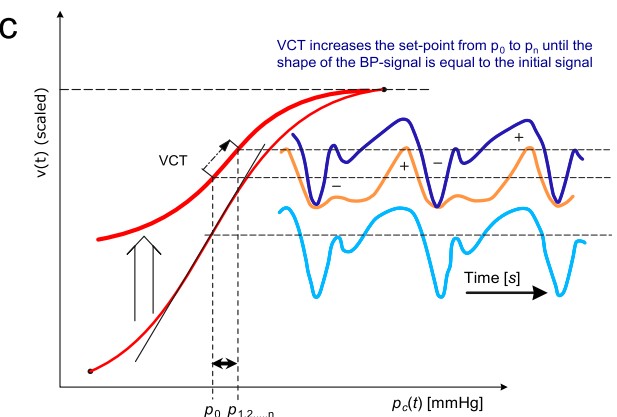

**Fig. 4 S-shaped *p-v* transfer function. a** The optimum pulsatile light signal $v_{Pulse}(t)$ (light blue trace) at optimal pressure $p_0$ has maximum amplitude with an integral over a full beat of zero. Suboptimal signals at too high ($p_>$) or too low ($p_<$) pressure are either spiky (dark blue trace) or fat (purple trace), respectively. mBP: mean blood pressure. **b** Vasoconstriction alters the S-shaped *p-v* transfer function and with it the pulse shapes of the light signals (from the light blue trace to the orange trace)–in particular the contributions of the positive ($+$) and negative ($−$) half-waves. **c** The setpoint $p_0$ and resulting $p_c(t)$ are altered after each heartbeat to $p_{1,2,...,n}$ using the volume control technique (VCT). In particular, the pulse shape is restored by VCT after vasoconstriction (see dark blue trace).

the CNAP HD technology allows the calculation of hemodynamic variables derived from the BP waveform.

CNAP2GO was implemented as software code (C++) and loaded as firmware on the CNAP Monitor HD. The final device was CE-marked as a class 2b device for clinical use in patients in

**Table 1 Participant characteristics and results of Bland Altman analysis for mean blood pressure.**

**Participant characteristics**

| Characteristic | Value |
| --- | --- |
| Male/female, *n* | 14/6 |
| Age, median (range) [years] | 36 (23–49) |
| Actual body weight, median (range) [kg] | 79 (56–130) |
| Height, median (range) [cm] | 177 (161–189) |
| CNAP Monitor HD sBP, median (range) [mmHg] | 108 (74–141) |
| CNAP Monitor HD dBP, median (range) [mmHg] | 64 (47–102) |
| CNAP Monitor HD mBP, median (range) [mmHg] | 82 (63–116) |

**Bland Altman analysis for mBP**

| Statistical quantity | Value |
| --- | --- |
| Mean of the differences±SD [mmHg] | 0.3 ± 4.4 |
| 95% limits of agreement [mmHg] | −8.3 to 8.9 |

sBP systolic blood pressure, dBP diastolic blood pressure, mBP mean blood pressure, SD standard deviation of the lab test series versus the CNAP Monitor. Total number of data points was $n = 400$ (i.e., 20 data points per participant).

the operating room. The sphygmomanometer of the CNAP Monitor HD was used for calibration.

**Lab tests versus CNAP Monitor HD**. Lab tests were performed in 20 healthy subjects (Table 1). In Fig. 5a and b, the BP and CO trends of the standard CNAP Monitor HD (VUT) and the CNAP2GO are shown, respectively. In order to provoke the cardiovascular system and induce changes in BP and vasomotor activity, the subjects performed several physiological maneuvers (deep breathing, fast breathing, submersion of contralateral hand into ice water, stroop test, passive leg raising) within 30 min.

The mean of the differences and its standard deviation between mBP measured with CNAP2GO and with CNAP Monitor HD were $0.3 ± 4.4$ mmHg.

Occasionally, unexpected motion artifacts appeared synchronously in BP signal streams of both the CNAP2GO and the CNAP Monitor HD. Movement only very briefly disturbed BP signal quality for both the CNAP2GO and the CNAP Monitor HD and the BP signal quality recovered immediately. Rapid signal recovery after signal alterations by finger movement is a characteristic feature of the existing CNAP method that we also implemented in the CNAP2GO method.

CO was calculated using the proprietary CNAP HD pulse wave analysis[39]. The bias between CNAP2GO CO and CNAP Monitor HD CO was $1.4 ± 0.6$ L/min with a percentage error (PE) of 22%. As shown in Fig. 5b, there was an inherent parallel shift in the CO trend. This may be caused by a different shape of BP pulse recorded with CNAP2GO which is directly derived from the superimposed PPG signal. The PE of 22% is below the 30% threshold that was defined by Critchley and Critchley to define clinically acceptable agreement between CO measurement devices[40]. The PE of 22% indicates that the CO trending information in the CNAP2GO BP signal is very similar to standard CNAP. This means that despite the nonlinearity of the vascular wall, the information in the PPG and BP waveform shapes are about the same, when contact pressure is at mBP and thus the artery is unloaded.

**Clinical study versus invasive reference method**. We included 46 patients having neurosurgery with general anesthesia in a clinical

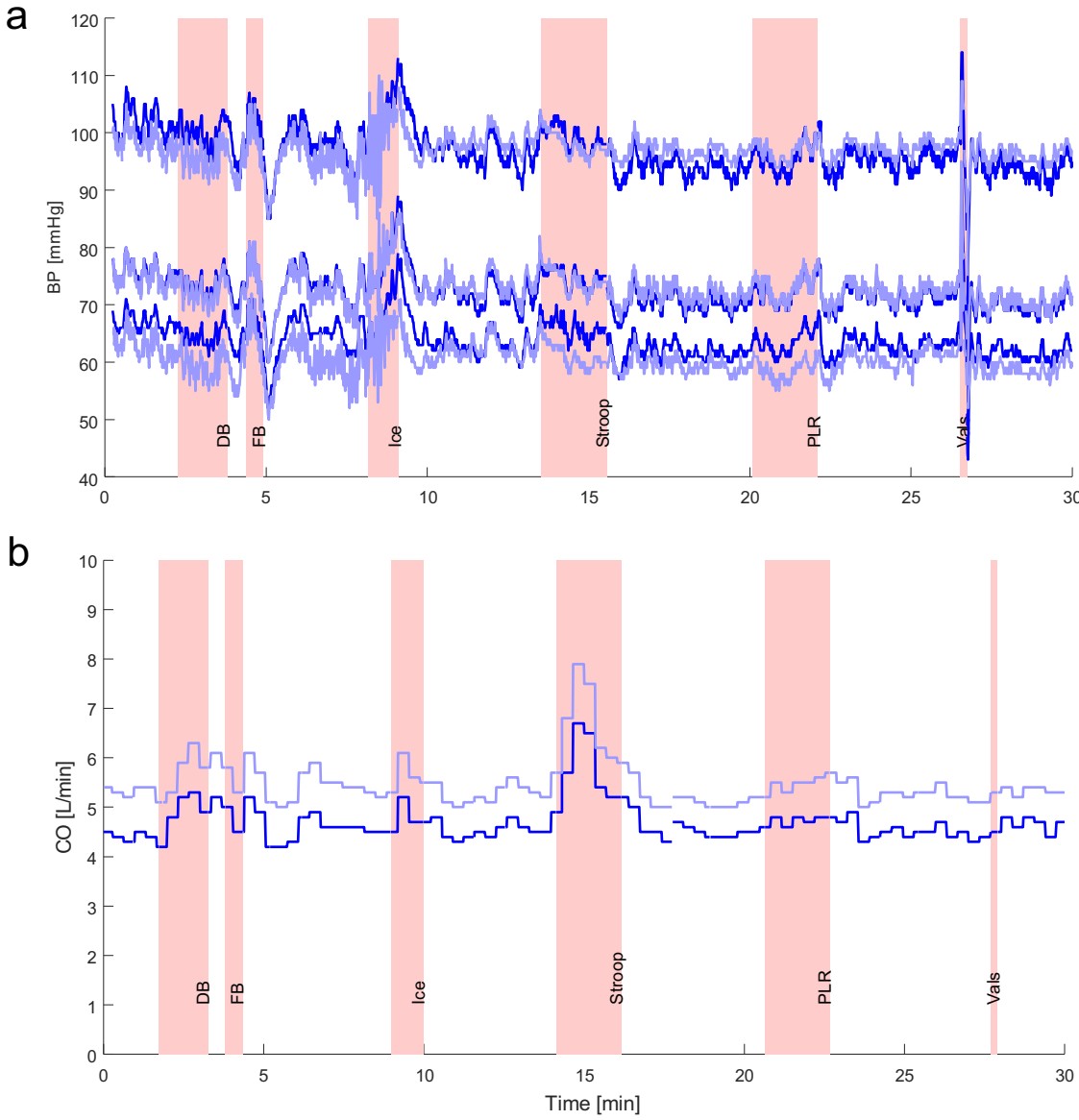

**Fig. 5 Sample blood pressure (BP) and cardiac output trend traces.** Data were taken from lab test measurements during physiological activities: "DB": deep breathing, "FB": fast breathing, "Ice": submersion of hand in ice water, "Stroop": color stroop test, "PLR": passive leg raising, "Vals": Valsalva test. **a** Beat-to-beat systolic, diastolic, and mean blood pressure values derived by CNAP Monitor HD (dark blue) and by CNAP2GO (light blue). **b** Cardiac output (CO) derived by CNAP Monitor HD (dark blue) and by CNAP2GO (light blue).

study. Invasive blood pressure (IBP) was recorded during routine care. We compared mBP measured using CNAP2GO and mBP measured using a radial arterial catheter (invasive reference method, clinical gold standard) during a period of about 45 min (Table 2). mBP ranged from 45 to 113 mmHg and thus covered hypo- as well as hypertensive phases. In the Bland Altman plot (Fig. 6a), we used color coding to illustrate measurements from the same patient, thus visualizing patient-specific offsets. Clustering indicates that patient-specific offsets remained similar throughout the recording which, in turn, indicates that CNAP2GO BP measurements were stable over the measurement period. The benchmark for acceptance was derived from the ISO 81060-2 (5 ± 8 mmHg). mBP from calibrated CNAP2GO and from the arterial catheter differed by −1.0 ± 7.0 mmHg over all patients.

When individually analyzing the data of each measurement, the mean of the differences for mBP ranged from −14.6 to 11.4 mmHg (interquartile range from −4.0 to 2.9 mmHg) with standard deviations ranging from 1.1 to 6.3 mmHg (interquartile range from 2.0 to 3.4 mmHg), indicating that the intra-subject difference to invasive reference was both stable as well as small for mBP values.

The rate of actuator pressure change, which according to VCT follows mBP, was analyzed in detail. The absolute changes in mBP of consecutive heartbeats ($n = 97{,}432$) were inspected together with the corresponding pulse intervals and the resulting need for actuator speed. The contact pressure of CNAP2GO required a median changing speed of about 1.4 mmHg per beat or 1.3 mmHg/s, the maximum values were 24.4 mmHg per beat or 25 mmHg/s.

Figure 7 shows the spectra of the contact pressure $p_c(t)$ in comparison with the frequency content of the whole pulsatile BP signal, demonstrating how much the control was slowed down for CNAP2GO.

**Wearable implementation.** The results of the clinical study reveal important findings for miniaturization of CNAP2GO.

Miniaturized CNAP2GO requires a small, low power actuator with a maximum speed of at most 25–30 mmHg/s, with an average speed of about 1.4 mmHg per beat. The actuator in interaction with the PPG system has to provide a correct oscillometric signal, needed for both the initial calibration and VCT principle. Here, correct means that the PPG signal has its origin only in the body segment and is not influenced by the mechanics of the sensor, e.g., a better signal coupling of the PPG elements due to higher actuator pressure. A constant signal-coupling over the complete pressure range can be achieved only with a homogenous pressure at the finger which underlines the essential role of sensor geometry.

Different prototypes were tested, the most promising system being a fluid-filled bladder containing the PPG elements in combination with a small motor compressing the fluid. Fluids can be incompressible gels or oils, for our experiment we used distilled water. Figure 8a shows a schematic of the actuator located in a finger-ring, Fig. 8b shows a 3D print of a possible CNAP2GO finger-ring.

Signal coupling of the PPG elements stayed constant with a $p_C(t)$ higher than a minimum contact pressure of about 20–40 mmHg. The fluid-filled bladder was located in a finger-ring. While we were focusing in this present prototype on the wearable sensor geometry, we used a small commercially available infusion pump as a fluid compressing driver.

We were able to demonstrate that this wearable fluid-filled bladder concept is able to produce correct oscillometric signals by testing the system within 7 subjects in comparison to the existing CNAP sensor. Results in comparison to a standard CNAP device are shown in Fig. 9a–n. Figures 10 and 11 show more details of the resulting oscillometric signals of subject 1.

**Important findings for the miniaturized actuator**. Drivers for actuators such as small printable piezoelectric motors are available with two important features: Low power consumption, but high enough stall torque to keep the actuator in its current position when no movement is needed. This has the effect that no energy is needed if mBP does not change.

We made basic calculations for a wearable CNAP2GO sensor using the data from a commercially available step motor (PCBMotor ApS, Ballerup, Denmark) to prove that the required pressure changes are feasible. We chose a printable motor with an outer diameter of 30 mm that can be integrated in the plate of a CNAP2GO finger-ring. This motor can change the volume in the fluid-filled bladder and thus contact pressure of the PPG elements via a gear from the motor. With a transmission ratio of the gear element higher than 20.27, this component has a high enough stall torque so that no energy is consumed as long as mBP is stable. With such a fluid-filled transmission system, the motor should theoretically change contact pressure for 1 mmHg within 10 ms. Assuming additional time due to control loops and further tolerances, we assume that a rate of pressure changes of up to 30 mmHg/s could be achieved with this particular piezo motor.

In miniaturized CNAP2GO, readjustments of $p_c(t)$ to follow mBP will be done as soon as the beat detector of the system has

**Table 2 Clinical study versus invasive reference method.**

**Patient characteristics**

| Characteristic | Value |
|---|---|
| Male/female, n | 19/27 |
| Age, median (range) [years] | 56 (25–79) |
| Actual body weight, median (range) [kg] | 77.5 (46–130) |
| Height, median (range) [cm] | 171.5 (157–192) |
| Invasive sBP, median (range) [mmHg] | 104 (60–172) |
| Invasive dBP, median (range) [mmHg] | 56 (29–103) |
| Invasive mBP, median (range) [mmHg] | 78 (48–131) |

**Bland Altman analysis for mBP**

| Statistical quantity | Value |
|---|---|
| Mean of the differences±SD [mmHg] | −1.0 ± 7.0 |
| 95% limits of agreement [mmHg] | −14.8 to 12.7 |

Patient characteristics and results of Bland Altman analysis for mean blood pressure. sBP systolic blood pressure, dBP diastolic blood pressure, mBP mean blood pressure, SD standard deviation of the clinical study versus invasive reference method. Total number of data points was n = 11,803.

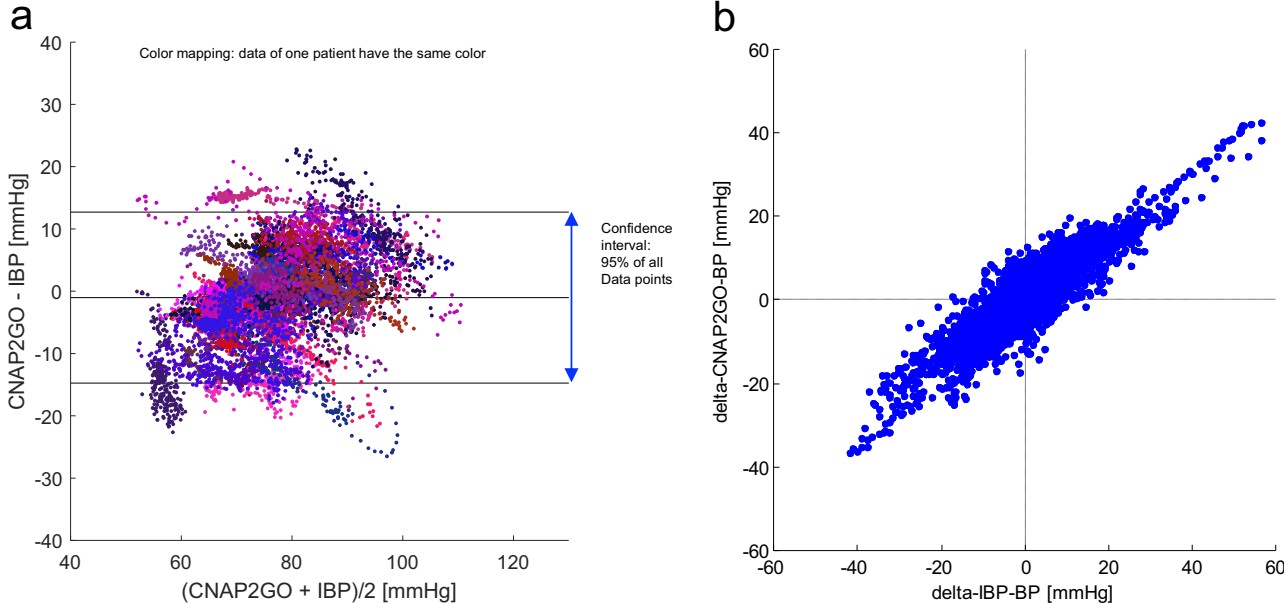

**Fig. 6 Results from clinical study. a** Bland Altman plot for mean blood pressure. Same-color points stem from the same patient. The plot includes indication of bias together with 95% limits of agreement which define the range in which 95% of all data points are expected to lie. Total number of data points was n = 11,803. **b** Concordance plot of spontaneous changes in mBP found within 5 min by invasive blood pressure (IBP) and CNAP2GO (i.e., delta-IBP-BP and delta-CNAP2GO-BP, respectively). Total number of changes assessed was n = 10,390.

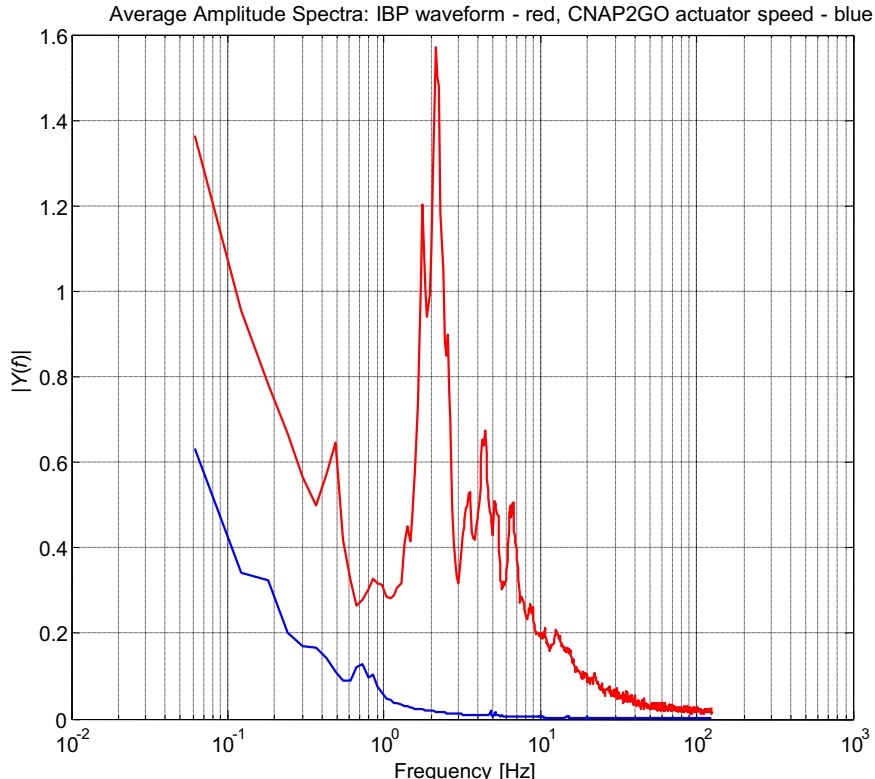

**Fig. 7 The spectra of CNAP2GO's contact pressure.** The frequency content of the contact pressure $p_c(t)$ (blue line) is shown in comparison to the frequency content of the whole pulsatile blood pressure signal (red line).

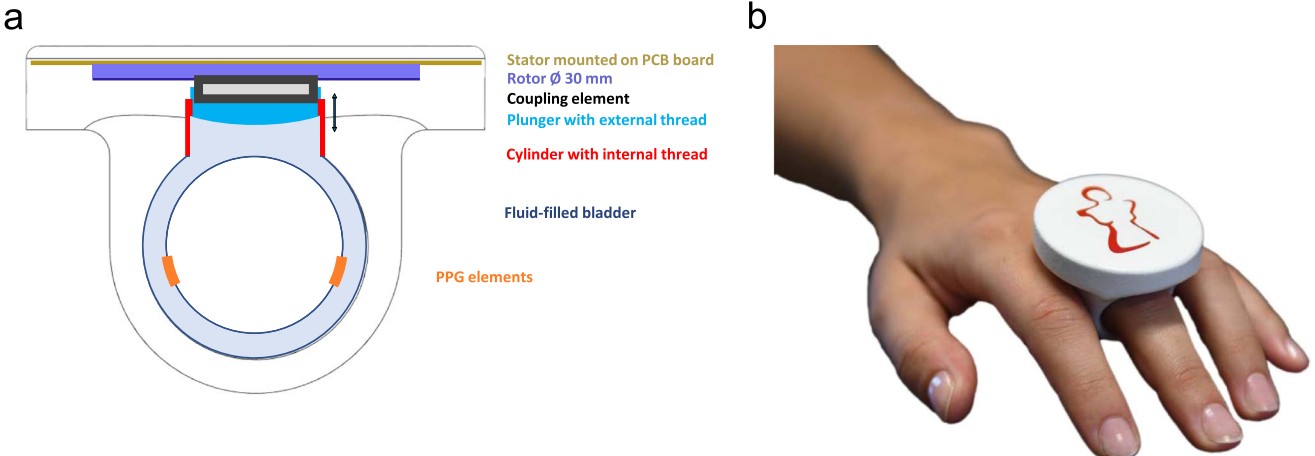

**Fig. 8 CNAP2GO finger-ring. a** Front view of the CNAP2GO finger-ring. The stator of the piezo motor is mounted on a printed circuit board (PCB). The rotor drives a plunger that is screwed into a cylinder, which presses on the fluid inside the bladder. The photoplethysmographic (PPG) elements record the light signal $v(t)$. **b** 3D print of a possible market-ready CNAP2GO finger-ring.

detected a heartbeat (usually about 100 ms after the systolic peak). Then the VCT criteria are used to derive the resulting new $p_c(t)$. In the validation experiments, $p_c(t)$ was adjusted using the CNAP pressure system and the finger cuff. In miniaturized CNAP2GO, the fluid-filled actuator will be activated until the new $p_c(t)$ is applied to the finger. Then the actuator will stop and keep $p_c(t)$ constant, at best, without power until the next heartbeat is detected.

An essential requirement for practical use and user acceptance is at least a 24 h use without battery reloading. We have performed estimations on the power consumption of CNAP2GO, with the following assumptions:

- Very significant BP changes of 2 mmHg per beat and a heart rate of 120 bpm
- Measurement phase including 45 s initial phase followed by 2 min of VCT-phase
- 13 min of low power interpolation mode (explanation next chapter)

The power needed to run the actuator will be 45.19 mW according to our calculation. Together with the PPG system (5.64 mW), a low-power bluetooth microcontroller (1.66 mW) and a motion sensor (6.21 mW), the power requirement of the CNAP2GO system is 58.7 mW or 1409 mWh during 24 h. For

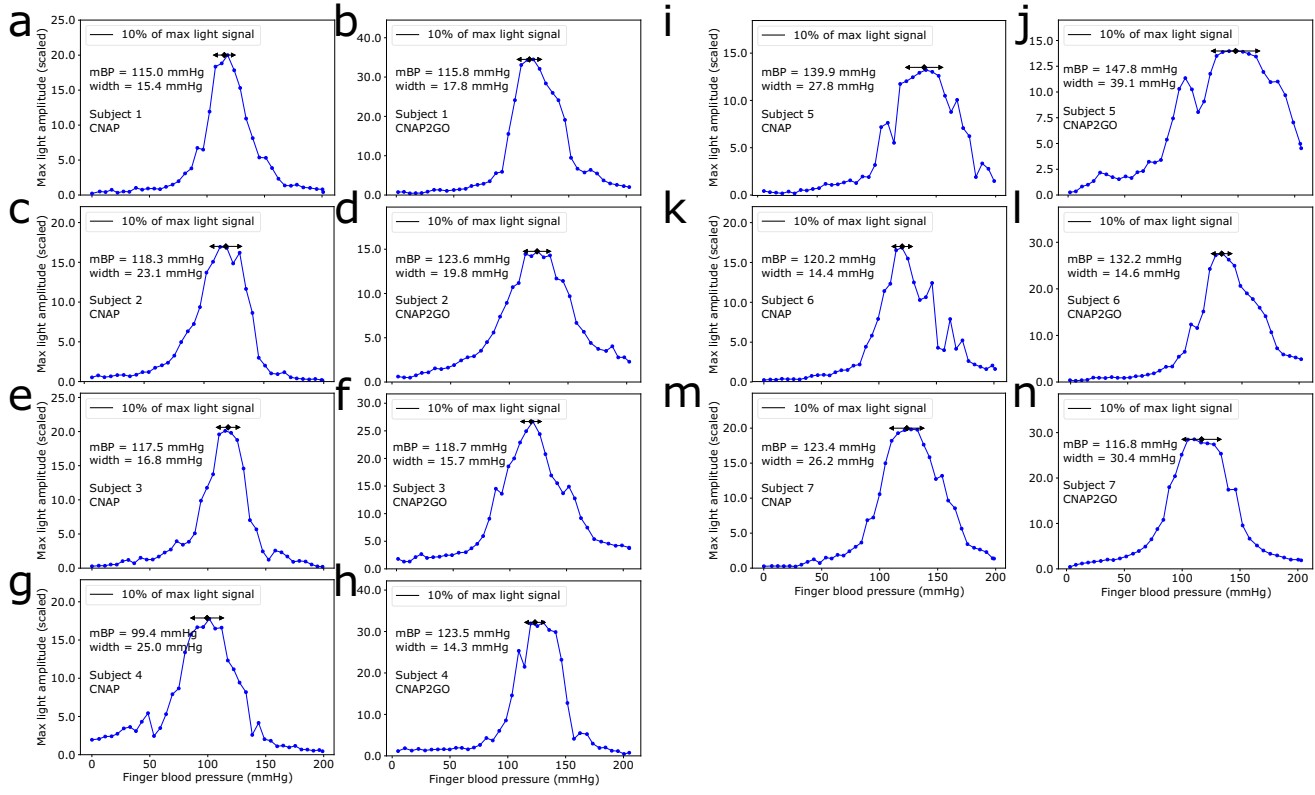

**Fig. 9 Oscillometric curves from 7 subjects obtained from the CNAP2GO prototype.** Compared are the results of the fluid-filled bladders in panels **a**, **c**, **e**, **g**, **i**, **k**, and **m** to the results of the standard CNAP with pneumatic control in panels **b**, **d**, **f**, **h**, **j**, **l**, and **n**, respectively. Mean blood pressure (mBP) and width of the Gaussian curves are denoted in each figure. The width is also demonstrated by black arrows. Measurement points (dots) are connected by straight lines.

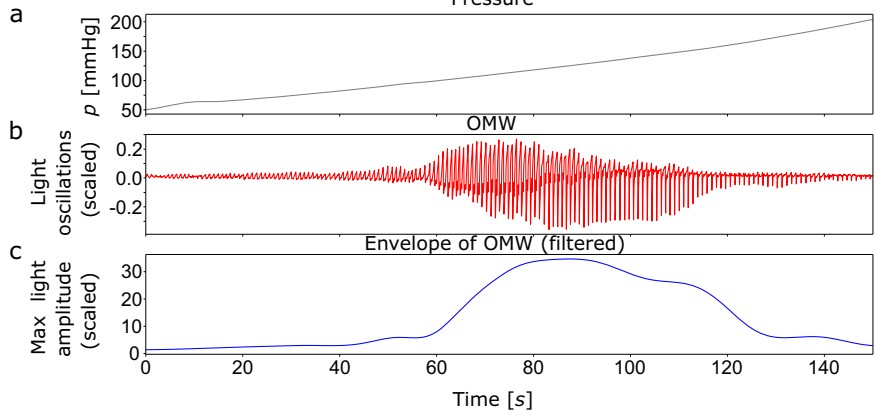

**Fig. 10 Detailed example of the oscillometric wave.** An example for an oscillometric wave (OMW) is shown from subject 1. **a** The pressure $p_c(t)$ is increased approximately linearly. **b** OMW – in detail $v_{Pulse}(t)$ – is recorded. **c** The resulting OMW envelope derived from the oscillations of $v_{Pulse}(t)$.

comparison, a single AAA-battery has 1500 mWh; when using two batteries or a rechargeable accumulator with more than 2000 mWh, a 24 h operation should be easily possible. As another example, the CNAP module, which is the core measurement unit without a user interface, has an average power consumption of 2.5 W with peaks up to 5 W.

**Further development challenges**. There are still some challenges regarding the implementation of this method in a wearable solution.

First, we did not perform our validation experiments using the final wearable hardware. Miniaturization will be a key task with

this sensor. Based on the present data set, all steps of miniaturization and technological refinement can be assessed and monitored in lab tests. Based on our tests, there is a well-founded assumption that the final CNAP2GO wearable can be located in a finger-ring as shown in Fig. 8b.

Second, for the wearable solution of CNAP2GO, the measured mBP needs a correcting element for orthostatic pressure whenever the finger level differs from the heart level. Heart level correction can be achieved using low-power motion sensors. Using small XYZ-axis accelerometers and gyroscopes, the orientation of human body segments and movements can be reconstructed and thus vertical difference ($d_v$) between the heart and the finger can be measured[41]. As the density of blood is

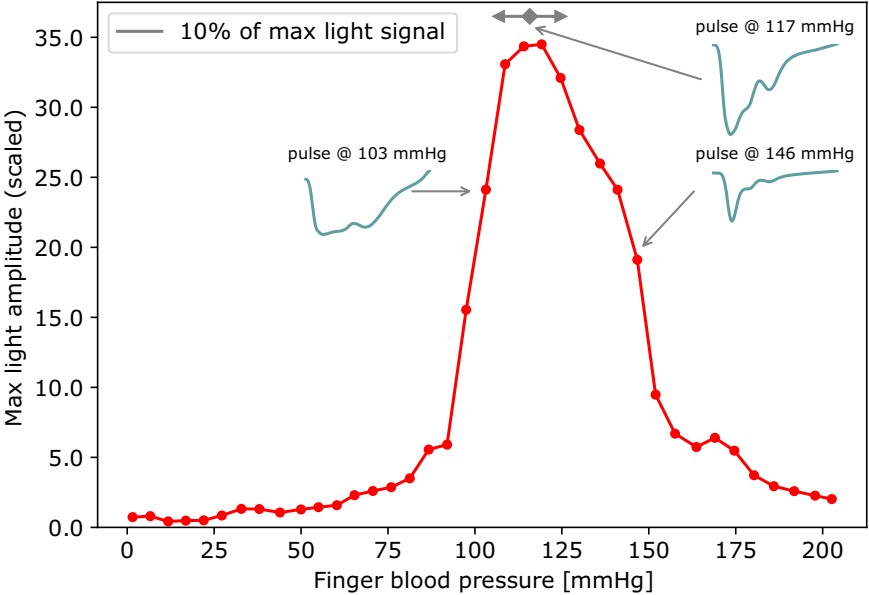

**Fig. 11 Shapes of the pulses.** The oscillometric wave envelope (red dots and lines) and typical fat, normal and spiky pulses (blue lines) from readings at defined pressure values as denoted in the figure. Measurement points (dots) are connected by lines.

known, hydrostatic pressure can be calculated from $d_v$ and simply subtracted from $p_C(t)$. Although the measurement of $d_v$ during motion needs further research and validation, this form of heart level correction is implementable as opposed to analog corrections for time-based wearable approaches, where the influence of posture and arm movement on, e.g., pulse transition is evident but solutions are unclear[42].

Third, the described CNAP2GO method still needs a continuous external pressure for correct measurements, but a 24 h continuous VCT measurement is clinically not necessary. Thus for better patient compliance, CNAP2GO will switch between a measurement mode and an interpolation mode. The measurement mode starts with the initial phase followed by the VCT-phase, where hemodynamic values are measured. These values will be fed into a mathematical model, which can be sufficiently trained after a short period of measurement (approximately 2 min). Afterward the external pressure can be reduced to a minimum, just high enough to ensure good light coupling. The trained mathematical model can then be used in the interpolation mode to estimate hemodynamic values from $v(t)$ or $v_{Pulse}(t)$ during this low pressure and low power consumption mode. The system will restart "measurement mode", either periodically or triggered by physiological changes (e.g., if prominent changes are detected in the morphology of the PPG signal, a restart of the "measurement mode" might be required). Figure 12 shows a simple flow diagram of these operation modes.

The purpose of the regularly occurring "measurement mode" phases is to record BP signals on the inflection point of the p-v transfer function. In this unloading state, the p-v transfer function is most linear allowing for the highest signal quality needed to derive advanced hemodynamic variables such as CO. This will assure clinical accuracy of BP and CO for the entire measurement.

The mechanism shown in Fig. 12 requires a sophisticated approach to switch between operation modes: The duration of the interpolation mode may be longer in stable hemodynamic phases (e.g., during sleep) and needs to be shorter in unstable situations, where enhanced measurement accuracy as guaranteed by the measurement phase is needed.

During the measurement phase of Fig. 12, the system will also be able to analyze variables of the autonomic system, e.g., baroreceptor reflex sensitivity and BP variability.

In patients with anticipated acute hypertensive or hypotensive episodes, two-finger rings may be used to avoid continuous external pressure on one finger. Two-finger rings may be used on contralateral hands or a double finger sensor—as already available for the standard CNAP system—on the same hand.

## Discussion

We have introduced the CNAP2GO method based on the art of VCT which requires only slowly changing actuators and is robust against vasomotor changes. We discovered that the pulsatile control in the standard VUT mainly avoids control system resonances. For CNAP2GO, we achieved the prevention of resonances by adaptive antiresonance filters.

We were able to demonstrate validity of CNAP2GO. mBP differed by $-1.0 \pm 7.0$ mmHg in comparison with the intra-arterial gold standard, which is well within the limits of $5 \pm 8$ mmHg demanded in the ISO 81060-2 standard for intermittent noninvasive sphygmomanometers[43].

Challenges of CNAP2GO are addressed in the article and reveal where further development has to be done:

- Hydrostatic correction between finger and heart level may be achieved by using small XYZ-axis accelerometers and gyroscopes.
- The lab tests show that CNA2GO is already resistant to motion artifacts. Artifacts may happen, but after stopping the movement, the system finds back to its correct trace. However, the system has not been tested during sport activities and other heavy movements.
- Miniaturization of the hardware components can be realized using PPG sensors with an LED and a photodiode. The contact pressure of the light elements to the skin is modifiable by a slow and thus low-energy actuator using a fluid-filled bladder, all of which make wearable CNAP2GO sensors feasible.
- A continuous external pressure has to be applied on the finger during VCT measurement. Figure 12 shows a mechanism

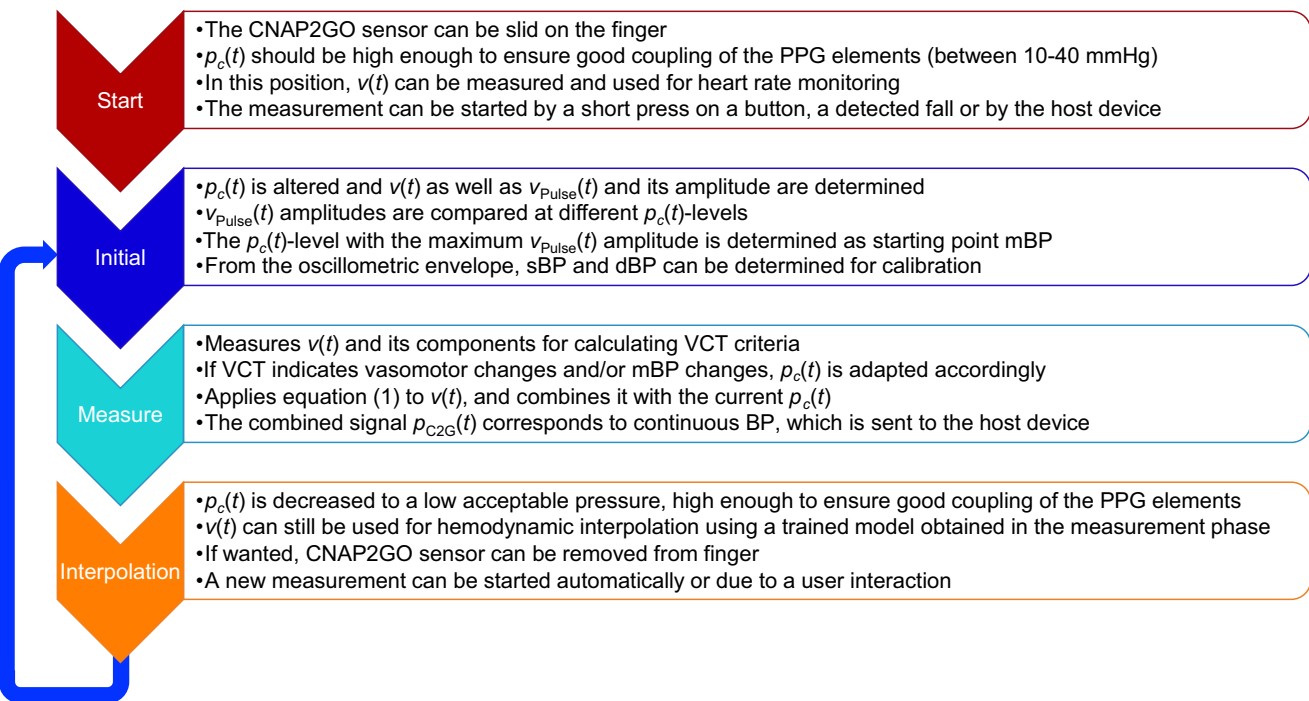

**Fig. 12 Software flow chart of the different CNAP2GO modes.** After the start mode, the system switches dynamically between initial, measure, and interpolation mode.

how the system may reduce the duration of pressure exposure by using an interpolation mode.

- CNAP2GO can be self-calibrated by an oscillometric measurement at the beginning of the measurement phase. Further validation work with the self-calibrated system needs to be done to prove clinical accuracy.

In conclusion, we were able to demonstrate that the CNAP2GO method enables mBP to be measured with clinically acceptable accuracy and precision, also during hypo- and hypertensive phases, in patients having surgery and subjects performing physiological maneuvers with hemodynamic activation. According to our experiments, this technique seems to be very well suited for wearable sensors integrated in a finger-ring, as shown in Fig. 8. The resulting continuous BP signal allows for the calculations of advanced hemodynamic variables such as CO, systemic peripheral resistance, baroreceptor reflex sensitivity, and many more. CNAP2GO potentially constitutes the breakthrough for wearable sensors for BP and blood flow monitoring in both ambulatory and in-hospital clinical settings.

## Methods

**Hardware**. To obtain a CNAP2GO prototype, we modified a standard CNAP Monitor HD, especially its core unit (the CNAP module). The CNAP module is an electronic system where almost all components are implemented as software on a 32-bit digital signal processor (DSP; TMS320F2810, Texas Instruments, Dallas, USA). Attached to the CNAP module is a double finger sensor system for alternating use on the index and middle finger. Each CNAP sensor contains a PPG system utilizing an LED ($\lambda = 890$ nm) as well as a light-to-frequency converter (TSL245R, ams, Unterpremstätten, Austria) for light detection that produces a digital pulse-width modulation signal. This signal goes directly to a timer input of the DSP producing the digital PPG (i.e., the time series v(t))[44]. The output of the DSP controls two valves (separate inlet and outlet valves). Both valves are arranged like transistors of a CMOS gate having one valve always closed. This allows for a high-fidelity contact pressure of the PPG elements in the finger sensor[45].

**Algorithm overview**. The original CNAP Monitor HD software in the module was replaced by the CNAP2GO/VCT method described in this article. All supporting functions (such as basic operating system, ambient light removal, valve controlling system, beat detector, data transmission, etc.) remained the same as in the original CNAP Monitor HD software V5.2.14.

**Initialization: Open-loop phase**. In the open-loop phase, a Gaussian-style oscillometric envelope is calculated by using the amplitudes of $v_{\text{Pulse}}$(t) obtained at different contact pressures $p_c$(t). The initial setpoint $P_0$ and its PPG counterpart $v_0$ are found according to the maximum amplitude rule. The filter variables of the cascades $v_{\text{Filt}}[1, 2,.., N]$ are then set to $v_0$.

**Digital signal filtering**. After ambient light removal, v(t) has a sampling frequency ($f_s$) of 250 Hz. A recursive digital IIR-filter was designed to obtain $v_{\text{Pulse}}$(t), $v_{\text{Rhythm}}$(t), $v_{\text{4Rhythm}}$(t), and $v_{\text{VCT}}$(t). In Fig. 13, frequency responses of the filters are shown, pseudocode of the filter cascade including coefficients are shown in Supplementary Method 1.

Each single filter stage contains a first-order IIR low-pass filter ($y_i = (1 - \text{UC}) * y_{i-1} + \text{UC} * x_i$). The recursive function provides down-sampling functionality to enable reasonable filter parameters for low cut-off frequencies.

After initialization in the open-loop phase, the filter adapts the relevant v-signals with every data point. Signals forget the initial setpoint $v_0$ with an update coefficient UC [1, 2,…, N].

**Reconstructing very slow changing BP changes**. In very slowly changing frequency components, blood volume and thus v(t) is not only influenced by BP but also by vasomotor activity. In the case of vasoconstriction, smooth vascular muscles close the arteries and arterioles, and finger blood volume decreases (i.e., v(t) goes up), although BP typically rises during vasoconstriction (i.e., v(t) should go down). The opposite behavior occurs during vasodilation. This means that the information from the PPG signal v(t) below $10^{-2}$ Hz is not reliable. Therefore, this frequency range is removed from all further calculations by filtering and using $v_{\text{VCT}}$(t).

Figure 13a and b show the frequency domain of the CNAP2GO control signals. Besides the high-pass filtered pulses $v_{\text{Pulse}}$(t), also the very-low-frequency components below $10^{-2}$ Hz of v(t) are removed from $v_{\text{VCT}}$(t) and $v_{\text{Rhythm}}$(t) by the digital filter design.

Within the VCT algorithm, pulses are inspected whether they have become spiky or fat within the previous beat by calculating the integral of $v_{\text{VCT}}$(t) over the actual beat. If the area over the negative half-wave differs from that over the positive half-wave, the integral over the complete beat is different from zero (Fig. 4b). Thus, the fat and spiky VCT control approach and its I summation reconstructs BP information below $10^{-2}$ Hz[46,47]. This then triggers an adjustment of the setpoint $P_n$ (Fig. 4c):

$$P_n = P_0 - c_{\text{BBI}} \cdot \sum_0^n \overline{V_n} - c_{\text{BBP}} \cdot \overline{V_n} \qquad (2)$$

where $c_{\text{BBI}}$ and $c_{\text{BBP}}$ are constants for the beat-based integral (BBI) and beat-based proportional (BBP) control approach, respectively. The continuous summation starting with the first beat allows for the reconstruction of the long-term BP information that is filtered away from $v_{\text{VCT}}$(t).

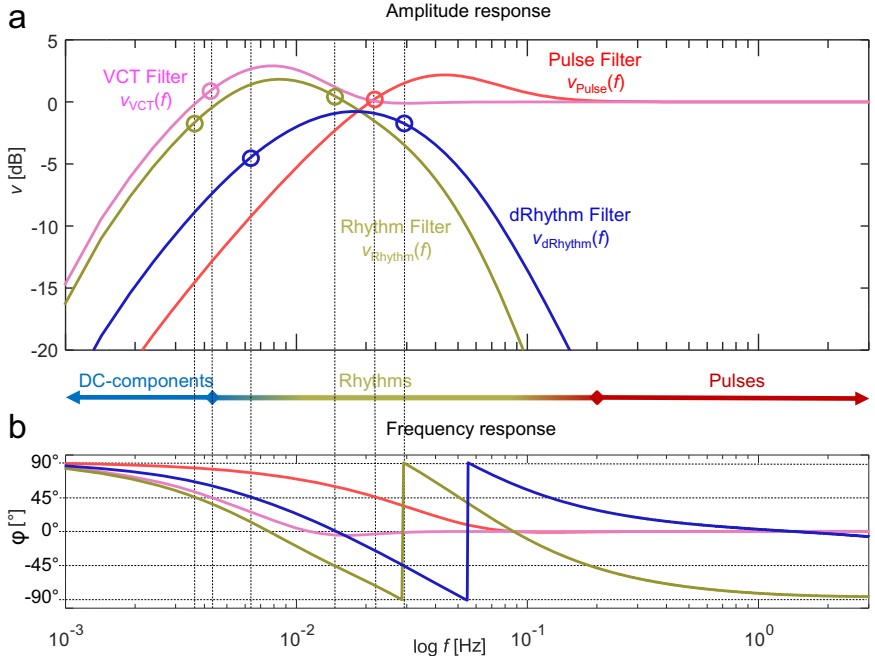

**Fig. 13 Control signals in the frequency domain – Bode diagram. a** The amplitude response of the volume control technique (VCT) control signals $v_{VCT}$, $v_{pulse}$, $v_{Rhythm}$, and $v_{dRhythm}$. Those are light signals filtered by the VCT, pulse, rhythm, and dRhythm filters, respectively. In particular, DC components, rhythms and pulses are discriminated. The dots and vertical lines represent the frequency cutoffs of each signal component. **b** Frequency response $\phi$ of the control signals. $f$: frequency.

The integral function for the beat is:

$$\overline{V_n} = \frac{1}{PI} \int_{t_{b-1}}^{t_b} v_{VCT}(t)dt \qquad (3)$$

where $t_b$ is the time t at which beat $b$ is detected and PI the pulse interval from $t_{b-1}$ to $t_b$.

**Tracking physiological rhythms**. VCT ensures a stable long-term tracking of mBP, corrects for vasomotor activity, and reconstructs slow-changing BP information. Theoretically, this mechanism should be a sufficient control signal, but it turned out that CNAP2GO did not follow physiological BP changes correctly. The control deviation of the system was too high and thus an additional mechanism for tracking fast mBP changes and physiologic BP rhythms was needed. Figure 13 shows $v_{Rhythm}(f)$, which covers physiological rhythms in BP that are related to breathing as well as to Traube–Hering–Mayer waves[48,49]. The signal $v_{Rhythm}(t)$ is led into a kind of a proportional-integral-differential (PID) control mechanism in order to keep $v_{Rhythm}(t)$ as small as possible by altering $p_c(t)$. Note that a rise in BP increases blood volume in the finger and thus decreases $v(t)$, whereas a fall in BP increases $v(t)$. The PID-control coefficients provoke negative feedback: if BP rises, $p_c(t)$ rises too while blood volume in the finger (indicated by $v_{Rhythm}(t)$) is kept constant.

The differential part of the controller, $v_{dRhythm}(f)$, takes care of pronounced BP changes (e.g., Valsalva maneuver, hyperventilation, blood loss, etc.). In order to follow the overall CNAP2GO principle, where actuator changes shall be slow, pulsatile blood volume is not clamped. Rather, the pulsatile frequencies are cut off, and the differential controller part acts as a second band-pass filter producing nonpulsatile $v_{dRhythm}(f)$.

Note: $v(f)$ indicates the frequency domain of the signal $v(t)$, as shown in Fig. 13. The CNAP2GO control system continuously keeps track of the required contact pressure with the following control structure:

$$y_{PD} = c_P \cdot v_{Rhythm} + c_D \cdot v_{dRhythm} \qquad (4)$$

$$y_I = c_I \cdot \sum_0^n y_{PD} \qquad (5)$$

The final CNAP2GO contact pressure is:

$$p_C(t) = P_n - y_I - y_{PD} \qquad (6)$$

where $c_P$, $c_D$, and $c_I$ are constants for the proportional, differential, and integral approach of the continuous control system, respectively, that keeps $p_c(t)$ at the inflection point of the S-curve and thus makes it equal to mBP.

**Antiresonance**. The complex control system outlined above follows the inflection point and thus mBP is tracked by $p_C(t)$. As a result, the speed of tracking is limited to the cut-off frequency of $v_{Rhythm}(f)$ and $v_{dRhythm}(f)$. This produces the control engineering problem that the stability of the system is reduced if the actuator is slow. Slow-reacting $p_C(t)$ creates a so-called pole in the z-plane indicating a tendency to oscillate. These oscillations especially occur when the patient calms down, parasympathetic tone increases, and the breathing frequency (about 0.2 Hz to 0.3 Hz) appears in the BP signal. A digital antiresonance system was developed that makes this pole ineffective. The basic considerations of this system were inspired by a method for suppressing mechanical resonance in high track density hard disk drives[50].

From CNAP2GO's need for antiresonance we see that a benefit of pulsatile VUT control systems is to inhibit this kind of resonance: Even in very fast pressure generators, pressure in the finger cuff will always lag behind true pulsatile BP. However, the resulting pole in a VUT control system is ineffective as long as the system can be faster than the BP signal.

Our antiresonance system inhibiting the resonances also for nonpulsatile control is thus a leap forward in the development of miniaturized BP measurement devices with acceptable clinical accuracy. It works as follows: The resonance detector indicates frequency and amplitude of a resonance oscillation in $p_c(t)$, which typically occurs between 0.1 Hz to 1 Hz. This information tunes a high-sensitive IIR notch filter which eliminates resonance phenomena. Pseudocode of the antiresonance system is shown in Supplementary Method 2.

**Lab tests versus CNAP Monitor HD**. Written informed consent was obtained from all participants. We performed measurements with the participants lying in the supine position. The finger sensor of the CNAP2GO system was attached to the same hand as the CNAP Monitor HD sensor. As usual for CNAP, both sensors were placed on the proximal phalanx. Both devices were initially calibrated to the same sBP/dBP values using the sphygmomanometer of the CNAP Monitor HD. Study measurements were performed over 30 min and included 6 physiological maneuvers.

For statistical analysis, simultaneously obtained mBP values measured with the CNAP2GO and with the CNAP Monitor HD were averaged over 10 s to reduce the number of comparisons and decorrelate consecutive data. We performed Bland Altman analysis accounting for multiple measurements per subject[51] and calculated the bias (mean of the differences between CNAP2GO mBP values minus CNAP Monitor HD mBP values), the standard deviation of the differences, and the 95% limits of agreement (Table 1).

**Clinical study versus invasive reference**. The clinical prospective method comparison study comparing mBP measured using CNAP2GO and mBP measured using a radial arterial catheter (invasive reference method, clinical gold standard)

was performed in the Department of Anesthesiology, Center of Anesthesiology and Intensive Care Medicine, University Medical Center Hamburg-Eppendorf (Hamburg, Germany) after approval by the ethics committee (Ethikkommission der Ärztekammer Hamburg, Hamburg, Germany). We obtained written informed consent from all patients. Adult patients scheduled for neurosurgical procedures in whom continuous BP monitoring with an invasive arterial catheter was planned, independently of the study, were eligible for study inclusion. Exclusion criteria were the presence of vascular abnormalities or anatomical deformities of the upper extremities or peripheral edema.

The finger sensor of the CNAP2GO system was attached to the hand opposite to the arterial catheter. CNAP2GO continuous noninvasive BP measurements and invasive reference measurements using a radial arterial catheter were recorded simultaneously over a period of 45 min. According to clinical standards, general anesthesia was maintained with remifentanil and propofol. Norepinephrine was given whenever clinically indicated.

Initial invasively measured BP values were used as post-hoc calibration values for CNAP2GO. For statistical analysis, mBP measured with CNAP2GO and mBP obtained using a radial arterial catheter were synchronized in time before being averaged over 10 s to reduce the number of comparisons and decorrelate consecutive data. We plotted mBP measured with the CNAP2GO system against invasively measured mBP values in scatter plots for visual assessment of the distribution and relationship of the BP data. In order to evaluate the agreement between mBP measured with CNAP2GO and mBP obtained using a radial arterial catheter we used Bland Altman analysis accounting for multiple measurements per patient[51] and calculated the bias (mean of the differences between CNAP2GO mBP values minus invasively assessed mBP values), the standard deviation of the differences, and the 95% limits of agreement (see Table 2 and Fig. 6a). Furthermore, we calculated the changes in BP (delta-BP) spontaneously occurring over a period of 5 min and investigated the concordance between changes in CNAP2GO mBP and changes in invasively measured reference mBP (see Fig. 6b).

**Lab tests with the fluid-filled bladder prototype**. Written informed consent was obtained from all participants. Measurements were performed with the subjects in a relaxed sitting position with the arms lightly resting on a table. The sensors of the prototype CNAP2GO and of the CNAP Monitor HD (used as reference) were placed on opposite hands. Measurements were performed simultaneously on index fingers, then simultaneously on middle fingers before switching the sensors between hands and repeating the procedure again on index and on middle fingers.

In order to obtain a finger oscillometric curve, finger cuff pressure was stepwise increased from 0 to 200 mmHg with approx. 2 mmHg/s. Cuff pressure was then plotted versus maximum light amplitude to visually compare the results obtained with the fluid-filled bladder prototype with the standard CNAP Monitor HD. While it was not expected that the prototype CNAP2GO would result in identical curves as the reference CNAP Monitor HD, the quality of the relationship between cuff pressure and maximum light amplitude was investigated: visual assessment found that the fluid-filled bladder driven by the slow actuator (infusion pump) provided oscillometric envelopes with a quality comparable to those provided by the CNAP Monitor HD (see Figs. 9, 10 and 11).

## Data availability

Data that support hypotheses, plots, and other findings of this study are available from the corresponding author upon reasonable request. The limited data insight has the background that contained data are sensitive personal data that were partially collected in a clinical study in patients. The data will only be passed on reasonable request and, according to the present study contract, if the recipient of the data agrees to comply with the European General Data Protection Regulation and not to attempt to trace the origin of the data.

## Code availability

Mathematical formulation of the detailed VCT is shown in the Methods. Pseudocode of VCT-related control algorithms like digital filters and the antiresonance system including coefficients are provided in Supplementary Methods 1-5. Code in Octave is available upon reasonable request. C++ code of the CNAP Monitor HD software V5.2.14 containing further supporting functions (such as basic operating system, ambient light removal, valve controlling system, beat detector, data transmission, etc.) is part of CNSystems' proprietary intellectual property and is not available for public use. Some of the methods are subject to different patents and patent applications.

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

## Acknowledgements
The study was financed by CNSystems Medizintechnik GmbH, Austria. A big part of the investment came from a crowdfunding initiative (https://www.lionrocket.com/cnsystems) and the authors thank all crowd investors. The authors thank Maximilian Rebhandl, who was engaged with building the prototypes and has performed the experiments within the 7 subjects resulting in Figs. 9, 10, and 11.

## Author contributions
J.F. developed the basic algorithms of CNAP2GO and supervised its development; he conceived and designed the study, was responsible for data interpretation, and drafted the manuscript. D.E.R. was responsible for data acquisition in the clinical study and data interpretation; she critically revised the manuscript for important intellectual content. C.F. was responsible for software implementation, he was responsible for data interpretation, and critically revised the manuscript for important intellectual content. D.F. was responsible for data analysis and data interpretation, and drafted the manuscript. J.G. contributed to the creation of the CNAP2GO software and was responsible for the development of the sensor; he was responsible for data interpretation and critically revised the manuscript for important intellectual content. K.L. was responsible for data analysis and data interpretation, and drafted the manuscript. B.S. conceived and designed the clinical study, was responsible for data interpretation, and drafted the manuscript. All authors approved the submitted version and agreed both to be personally accountable for the author's own contributions and to ensure that questions related to the accuracy or integrity of any part of the work, even ones in which the author was not personally involved, are appropriately investigated, resolved, and the resolution documented in the literature.

## Competing interests
CNSystems Medizintechnik GmbH (Graz, Austria) develops, manufactures, and markets the noninvasive CNAP technology. J.F. is CEO and founder of CNSystems, receives salary, and has equity interests. C.F., D.F., J.G., and K.L. are employees of CNSystems. J.F., C.F., J.G., and K.L. are inventors and named on one or more patents regarding continuous noninvasive technologies. B.S. has received institutional restricted research grants, honoraria for giving lectures, and refunds of travel expenses from CNSystems. D.E.R. has no conflicts to declare.
