## [Peer Review File · Nature Communications]

Reviewer #1 (Remarks to the Author):

This manuscript by Fortin et al. reported a recent effort from the team to modify their existing commercially available CNAP Monitor HD to monitor blood pressure from a finger via plethysmography (PPG) signals by adding a feedback loop and an actuator, aiming to establish a novel volume control technique mechanism. The system was tested with 20 healthy individuals and then validated with 46 patients with arterial line blood pressure measurement.

The manuscript was pretty well written with a lot of technical details though it was a little bit hard to follow all the description and explanation. The statistical results were well presented using the Bland Altman plot, indicating a promising technique. The novelty of the entire work lies in the closed-loop volume control technique.

Here are some concerns.

Similar to many other reports using PPG or PPG+some other signals to derive BP, the system still needs calibration! This really limits the practical use, especially for diagnosis purpose in the home setting. Although the authors have discussed this, it still dampened the enthusiasm of audience. Further, due to the indirect approach to derive BP, there could be many uncertainties, thus the results have been found to vary from person to person and time to time. Thus, for validation, it may be better to conduct several experiments for the same subjects in different days. To present the data and investigate the issue, the authors may look at outliers who gave worst accuracy.

The authors claim that their newly modified system can be used for wearables with miniaturization. Of course, it can be. However, how does it differentiate from others, even with the old one? More tests will need to be conducted in order to confirm its superior features. For example, there should be a line of experiments in which motion artifacts are rigorously investigated and multiple devices/approaches are used simultaneously for comparison.

For clinical use, again outlier cases need to be pulled out to discuss and study.

Minors: Section 2 was named Results though the content below were not about results. Line 254: type "raining". Table 1: The word patient was used.

Reviewer #2 (Remarks to the Author):

This manuscript presents a new art of continuous non-invasive arterial blood pressure monitoring (CNAP2GO) and shows excellent BP measurement performance in comparison with invasive BP in 46 patients having surgery. I think this manuscript is quite interesting and have a large impact. If the method in this manuscript is in practical use, the concept of health care would be changed dramatically. From the point of accuracy, the measurement accuracy of CNAP2GO is verified in this manuscript. I think this manuscript is worth being accepted, however, I have a few questions, thus I wish the authors would answer my questions.

1. The authors say that the power requirement of the CNAP2GO system is less than 60 mW. This is quite important point for practical use and mass production. However, only the value of 60 mW is a bit difficult to conceive. Could you show comparison the 60 mW (i.e. the power requirement of the

CNAP2GO) and other electrical devices? If the authors would show this, the utility of CNAP2GO could be appealed more.

2. Considering the final wearable hardware, could the authors show the total size of CNAP2GO? It's okay with the size the authors currently assume. I think the compactness could be one of the appeal points.

Reviewer #3 (Remarks to the Author):

The authors modified the volume clamping technique to measure mean BP and proved the accuracy of the technique against a-line measurements in neurosurgical patients. The main point is that the modified technique clamps only the mean of the PPG and therefore requires much slower actuation. This allows the technique to be more amenable to implementation as a ring worn on the finger. I personally found the paper to be interesting and did not find any conceptual flaws in the technique.

It makes sense to first use existing stationary and expensive hardware to test the technique. But to capture the interest of this journal's broad readership, I believe some kind of wearable implementation is necessary.

A wearable implementation is nontrivial from miniaturization and power perspectives and otherwise. The authors mention the important hydrostatic issue which can cause BP to change by +/-50 mmHg without any change in systemic pressure. But I believe they understate this point. The degrees of freedom for a finger ring sensor are the elbow, wrist, and finger joints. This makes using accelerometers difficult to detect the distance to the heart. A fluid-filled tube with a pressure sensor could be attached to the ring from the arm. But this would likely not be adopted outside of hospitals. Another complexity is motion artifact which the authors did not mention.

Wearable implementation aside, a key disadvantage of the proposed technique is the need for continual external pressure application at the mean BP. This is the same on average as conventional volume clamping. Compounding matters finger cuffs are placed at the middle of the finger whereas a ring would be worn at the finger base. Conventional volume clamp devices also use two finger cuffs to reduce this issue. Long term tolerance for the wearable ring is therefore questionable.

I agree that accuracy is currently a challenge for timing techniques such as pulse transition time. But the advantage of these techniques is that the sensors can be placed at much lower external pressure. This was not mentioned in the paper.

Also there are other devices that were not mentioned. Most notable is the Omron HeartGuide watch BP monitor which is FDA cleared. The advantage of the proposed technique is that it is continuous. At the same time continuous could be disadvantage. The Omron watch makes on-demand measurements. The watch is held at heart level during these measurements. Therefore hydrostatic issues and motion artifact are not a factor as is continual external pressure application. Spot measurements of absolute BP have also been proposed with smartphones (see recent Chandrasekhar papers).

The authors slow down the actuation and therefore sacrifice direct measurement of systolic and diastolic BP. They estimate these quantities by assuming the PPG and BP waveform shapes are same when the artery is unloaded. But the finger artery is highly nonlinear and viscoelastic. In fact the cardiac outputs differed appreciably between conventional volume clamping and the modified volume clamping methods.

It may make more sense to periodically employ oscillometry to directly measure systolic and diastolic BP every 5 to 10 min. This would also limit the external pressure application. In this case the advantage of the proposed technique would only be to identify faster changes in mean BP (within 5 to 10 min).

It is not clear to me why $v_{\{VCT\}}$ is not enough for the control signal? For instance the PPG amplitude/shape should change with respiratory BP waves. Please explain why $v_{\{Rhythm\}}$ and $v_{\{dRhythm\}}$ are needed on top of $v_{\{VCT\}}$. Also please describe the technique so that it can be reproduced including information such as coefficient values.

There is discussion of calibration in the paper. Does this refer to transforming finger pressure to arm pressure? Please clarify. Perhaps finger pressure would suffice for a hospital application of rapid hypotension detection postsurgically? This is not as exciting an application as hypertension but is important nonetheless.

Point-by-point response

REVIEWER 1:

REVIEWER COMMENT #1: Similar to many other reports using PPG or PPG+some other signals to derive BP, the system still needs calibration! This really limits the practical use, especially for diagnosis purpose in the home setting. Although the authors have discussed this, it still dampened the enthusiasm of audience.

RESPONSE: Thank you for this comment that showed us that we did not describe clearly enough that the proposed method can self-calibrate. We now describe in more detail how the CNAP2GO takes oscillometric readings at the finger that can be used for calibration and also for finding the initial set-point. We also included more references. Please note that we also added a comment about existing “Vascular Unloading Technique” (VUT) devices. The Finapres family does not need calibration, but a transfer function to brachial BP-values. Other transfer functions are also described in the literature. For CNAP2GO, we also will use a transfer function.

REVIEWER COMMENT #2: Further, due to the indirect approach to derive BP, there could be many uncertainties, thus the results have been found to vary from person to person and time to time. Thus, for validation, it may be better to conduct several experiments for the same subjects in different days. To present the data and investigate the issue, the authors may look at outliers who gave worst accuracy.

RESPONSE: All international standards including the upcoming ISO 81060-3 for continuous BP devices do not recommend inter-patient/inter-subject validation methods. Thus, we focused on clinical data from 46 patients having surgery and 20 subjects performing physiological maneuvers influencing hemodynamics. We did not exclude patients or subjects from the final validation. If outliers would play a

significant role, agreement of validation (0.3 ± 4.4 mmHg in lab-tests and 1 ± 7 mmHg in clinical data) would be worse.

REVIEWER COMMENT #3: The authors claim that their newly modified system can be used for wearables with miniaturization. Of course, it can be. However, how does it differentiate from others, even with the old one? More tests will need to be conducted in order to confirm its superior features.

RESPONSE: This is an important comment helping us to substantially improve the manuscript. For this revised version, we have built many prototypes and have performed promising tests, which we now describe in the manuscript under the subheading “Wearable implementation”. We thus show that the actuator principle provides reliable oscillometric curves which are the basis for “Volume Control Technique” (VCT) as well as for self-calibration. The summarize findings of additional experiments demonstrating the feasibility of miniaturizing the system (“Important findings for the miniaturized actuator”) and provide a critical discussion. The energy calculations (see discussion) have been reworked, although there was no significant change in the prospective power consumption.

REVIEWER COMMENT #4: For example, there should be a line of experiments in which motion artifacts are rigorously investigated and multiple devices/approaches are used simultaneously for comparison.

RESPONSE: We now cover this important aspect in the revised manuscript. We continuously recorded blood pressure for more than 10 hours in the lab test and for about 45 minutes per patient in the operating room. Expectedly, we observed many unexpected motion artifacts during data recording. Motion artifacts can affect all

measurement methods and can thus be observed in all signals: CNAP2GO, reference CNAP, and invasive (reference) blood pressure signal.

During lab tests, movement of the fingers in the sensors have been recorded on video thus allowing us to detect and document movement artifacts. Movement only very briefly disturbed blood pressure signal quality for both the CNAP and CNAP2GO method and the blood pressure signal quality recovered immediately. Rapid signal recovery after signal alterations by finger movement is a characteristic feature of the existing CNAP method that we also implemented in the CNAP2GO method.

REVIEWER COMMENT #5: For clinical use, again outlier cases need to be pulled out to discuss and study.

RESPONSE: We agree that thoroughly reviewing and discussing outlier cases is important when validating a measurement system for clinical use. Therefore, more validation data in different clinical settings will be required for the final miniaturized device. For the current study, all patient recordings were used for validation – no patients have been excluded. When looking at the results of each individual recording, the mean of the differences for mBP ranged from -14.6 to 11.4 mmHg (interquartile range from -4.0 to 2.9 mmHg). The standard deviation of the intra-subject differences between CNAP2GO and invasive reference blood pressure measurements ranged from 1.1 to 6.3 mmHg for mBP (interquartile range; 2.0 to 3.4 mmHg) and thus provides an indication that the differences vary only slightly within each measurement.

REVIEWER COMMENT #6: Minors: Section 2 was named Results though the content below were not about results. Line 254: type "raining". Table 1: The word patient was used.

RESPONSE: Changed and corrected.

REVIEWER 2:

REVIEWER COMMENT #1: The authors say that the power requirement of the CNAP2GO system is less than 60 mW. This is quite important point for practical use and mass production. However, only the value of 60 mW is a bit difficult to conceive. Could you show comparison the 60 mW (i.e. the power requirement of the CNAP2GO) and other electrical devices? If the authors would show this, the utility of CNAP2GO could be appealed more.

RESPONSE: This is an important comment. We revised the power requirement section and moved it to the discussion section. We added that 1409 mWh are needed to use the system for 24 hours. For comparison, a single AAA-battery provides 1500 mWh, when using two batteries or a rechargeable accumulator with more than 2000mWh, using the system for 24 hours is easily possible. For comparison, the CNAP module, which is the core measurement unit without user interface, has an average power consumption of 2.5W with peaks up to 5W.

REVIEWER COMMENT #2: Considering the final wearable hardware, could the authors show the total size of CNAP2 GO? It's okay with the size the authors currently assume. I think the compactness could be one of the appeal points.

RESPONSE: In Figure 6b we show a picture of a (non-functioning) 3D-print of the envisioned final product.

REVIEWER 3:

REVIEWER COMMENT #1: It makes sense to first use existing stationary and expensive hardware to test the technique. But to capture the interest of this journal's broad readership, I believe some kind of wearable implementation is necessary.

RESPONSE: As can be seen in our reply to Reviewer 1, Comment #3, we have built and tested new prototypes and have performed validation tests by inspecting the resulting oscillometric curve of the new actuator/sensor.

REVIEWER COMMENT #2: A wearable implementation is nontrivial from miniaturization and power perspectives and otherwise. The authors mention the important hydrostatic issue which can cause BP to change by +/-50 mmHg without any change in systemic pressure. But I believe they understate this point. The degrees of freedom for a finger ring sensor are the elbow, wrist, and finger joints. This makes using accelerometers difficult to detect the distance to the heart. A fluid-filled tube with a pressure sensor could be attached to the ring from the arm. But this would likely not be adopted outside of hospitals.

RESPONSE: Thank you for this important point. We have added more details including the important reference from Luinge and Veltink (Ref 45). They use small XYZ-axis accelerometers and gyroscopes for the measurement of orientation of human body segments and movement. In the introduction of this article, one can find many other references where accelerometers and gyroscopes are used in medical applications.

REVIEWER COMMENT #3: Another complexity is motion artifact which the authors did not mention.

RESPONSE: We have added more information on motion artifacts – please see our response to Reviewer 1, Comments #2, #4, and #5.

REVIEWER COMMENT #4:

Wearable implementation aside, a key disadvantage of the proposed technique is the need for continual external pressure application at the mean BP. This is the same on average as conventional volume clamping.

RESPONSE: We appreciate these thoughtful comments and revised the manuscript by adding our idea of the “Interpolation Mode” (see discussion and Figure 7). In a basic version we have described the different modes of CNAP2GO in Ref. 52.

REVIEWER COMMENT #5: Compounding matters finger cuffs are placed at the middle of the finger whereas a ring would be worn at the finger base. Conventional volume clamp devices also use two finger cuffs to reduce this issue. Long term tolerance for the wearable ring is therefore questionable.

RESPONSE: Conventional CNAP devices such as the CNAP Monitor 500, CNAP Modules for Dräger (Lübeck, Germany) and LiDCO (London, UK), NiCCI (Getinge, Solna, Sweden) as well as the Task Force Monitor (CNSystems) use the proximal finger phalanx. All of these devices use a double finger sensor to change from one finger to the other. CNAP2GO shall change between measurement mode and interpolation mode according to Figure. 7.

REVIEWER COMMENT #6: I agree that accuracy is currently a challenge for timing techniques such as pulse transition time. But the advantage of these techniques is

that the sensors can be placed at much lower external pressure. This was not mentioned in the paper.

RESPONSE: As mentioned in the responses to comments #4 and #5, we have added the interpolation mode as described in the discussion and Figure 7.

REVIEWER COMMENT #7: Also there are other devices that were not mentioned. Most notable is the Omron HeartGuide watch BP monitor which is FDA cleared. The advantage of the proposed technique is that it is continuous. At the same time continuous could be disadvantage. The Omron watch makes on-demand measurements. The watch is held at heart level during these measurements. Therefore hydrostatic issues and motion artifact are not a factor as is continual external pressure application. Spot measurements of absolute BP have also been proposed with smartphones (see recent Chandrasekhar papers).

RESPONSE: Thank you for this important comment. We have mentioned the HeartGuide and Chandrasekhar's smartphone approach, which we really find highly innovative. Chandrasekhar et al also show, that finger oscillometry works. Both methods need user interaction, whereas the final CNAP2GO will work independently.

REVIEWER COMMENT #8: The authors slow down the actuation and therefore sacrifice direct measurement of systolic and diastolic BP. They estimate these quantities by assuming the PPG and BP waveform shapes are same when the artery is unloaded. But the finger artery is highly nonlinear and viscoelastic. In fact the cardiac outputs differed appreciably between conventional volume clamping and the modified volume clamping methods.

RESPONSE: Especially the CO-results show that PPG and BP carry the same information when the artery is unloaded, although the signals look different. CO

comparison of the lab tests show, that the constant bias is high (1.4 L/min), but the standard deviation (0.6 L/min) resulting in a PE of 22% is in a relatively low range according to Critchley and Critchley. This means that despite the non-linearity of the vascular wall, the PPG- and BP-waveform shapes are about same and allow for a good trending ability of the method, when contact pressure is at mean BP and thus the artery is unloaded.

REVIEWER COMMENT #9: It may make more sense to periodically employ oscillometry to directly measure systolic and diastolic BP every 5 to 10 min. This would also limit the external pressure application. In this case the advantage of the proposed technique would only be to identify faster changes in mean BP (within 5 to 10 min).

RESPONSE: The advantage is the automatic measurement of hemodynamics without user interaction. For cardiac output and further derived hemodynamic values, a continuous BP signal is needed. In addition, for an assessment of autonomic function, physiological rhythms (Traube-Hering-Mayer waves, breathing rhythms) need to be investigated. For the use for patient monitoring in low-acuity settings, all BP changes such as hypotensive phases need to be monitored.

REVIEWER COMMENT #10: It is not clear to me why $v_{\{VCT\}}$ is not enough for the control signal? For instance the PPG amplitude/shape should change with respiratory BP waves. Please explain why $v_{\{Rhythm\}}$ and $v_{\{dRhythm\}}$ are needed on top of $v_{\{VCT\}}$.

RESPONSE: VCT alone is not good enough for tracking physiological rhythms. Thus, $v_{\{Rhythm\}}$ and $v_{\{dRhythm\}}$ are needed to increase accuracy in this frequency range. We now describe this in the revised methods section.

REVIEWER COMMENT #11: Also please describe the technique so that it can be reproduced including information such as coefficient values.

RESPONSE: We added supplementary material and disclose coefficient values as well as pseudocode of supporting algorithms (digital filters, anti-resonance).

REVIEWER COMMENT #12: There is discussion of calibration in the paper. Does this refer to transforming finger pressure to arm pressure? Please clarify.

RESPONSE: We have added more details on the calibration obtained from the initial finger oscillometric curve. Please also see our response to Reviewer #1 comment #1.

REVIEWER COMMENT #13: Perhaps finger pressure would suffice for a hospital application of rapid hypotension detection postsurgically? This is not as exciting an application as hypertension but is important nonetheless.

RESPONSE: Our validation data show, the hypotensive and hypertensive phases are correctly measured with CNAP2GO. We agree that postoperative hypotension is a major medical problem. Postoperative hypotension during the initial days after surgery is common and is independently associated with postoperative myocardial injury and a composite of myocardial infarction and death, even after adjusting for intraoperative hypotension. Therefore, using CNAP2GO for early recognition of hypotension occurring after surgery on the normal ward is an intriguing application. We mention this in the revised manuscript.

Reviewer #1 (Remarks to the Author):

The authors have addressed all comments carefully and performed a thorough revision.

Reviewer #2 (Remarks to the Author):

Thank you for resubmitting your manuscript. I read through your resubmitted manuscript, and It is found that the manuscript was revised moderately. I think this manuscript is worth being published.

Reviewer #3 (Remarks to the Author):

I would like to thank the authors for being so responsive to my initial comments.

I appreciate Figure 7. But, I don't see how this would be any better than periodically performing oscillometry. Volume clamping may not be necessary to compute cardiac output as well. I also think that Figure 7 takes away from the study.

The main advantage of the authors' method is that it can permit *continuous* monitoring of MAP in a principled way and in potentially a ring form factor.

The main disadvantage is the necessary continual external pressure on the finger. But, you could use two rings, eg, one on each hand. This ring set could be used for special patients such as those with acute hypertensive or hypotensive episodes.

I think the paper would be better if you just clearly stated the main advantage and disadvantage.

Major challenges, which I believe are still understated, are hydrostatic effects, motion artifact, and achieving the actual ring form factor.

As a minor point, PPG at zero internal minus external pressure is where the artery is most nonlinear.

As a final important point, it appears that the pseudo code may not be complete. Could you include all pseudo code including the control mechanism?

Point-by-point response

REVIEWER 3:

COMMENT 1: I appreciate Figure 7. But, I don't see how this would be any better than periodically performing oscillometry. Volume clamping may not be necessary to compute cardiac output as well. I also think that Figure 7 takes away from the study.

RESPONSE:

Thank you for your comment. We added Figure 7 to the manuscript to illustrate the concept of operation modes and their fundamental functionality. In the revised manuscript, we now additionally provided further explanations on why we consider the continuous measurement mode using VCT to be substantial for the accuracy of our wearable approach.

COMMENT 2: The main advantage of the authors' method is that it can permit *continuous* monitoring of MAP in a principled way and in potentially a ring form. The main disadvantage is the necessary continual external pressure on the finger. But, you could use two rings, eg, one on each hand. This ring set could be used for special patients such as those with acute hypertensive or hypotensive episodes. I think the paper would be better if you just clearly stated the main advantage and disadvantage.

RESPONSE: This is an excellent comment. In the revised manuscript, we now describe clearly the advantages and disadvantages of the method in the discussion section. In addition, we discuss a two ring setup, which is similar to the double finger concept of the CNAP clinical device.

COMMENT 3: Major challenges, which I believe are still understated, are hydrostatic effects, motion artifact, and achieving the actual ring form factor.

RESPONSE: According to this valuable comment we expanded on these challenges in the limitations section of the revised manuscript.

COMMENT 4: As a minor point, PPG at zero internal minus external pressure is where the artery is most nonlinear.

RESPONSE: If this comment implies “zero (internal minus external pressure)”, we respectfully disagree. When external pressure equals internal pressure the artery is in an “unloaded” state and is most linear – see figure 4a at pressure p_0 . VCT has the in-built mechanism of keeping the artery in this unloaded linear state.

COMMENT 5: As a final important point, it appears that the pseudo code may not be complete. Could you include all pseudo code including the control mechanism?

RESPONSE: In addition to the detailed description of the control mechanisms in the methods section, we added a “main routine” of the control mechanism to the pseudo code.

Reviewer #3 (Remarks to the Author):

Thank you for your thorough responses. This is a good contribution to the field.